# Molecular mechanisms of splenectomy-induced hepatocyte proliferation

**Andrey V. Elchaninov**[1,2]*, **Timur Kh. Fatkhudinov**[2,4], **Polina A. Vishnyakova**[1], **Maria P. Nikitina**[4], **Anastasiya V. Lokhonina**[1], **Andrey V. Makarov**[1,3], **Irina V. Arutyunyan**[1], **Evgeniya Y. Kananykhina**[4], **Anastasiya S. Poltavets**[1], **Kirill R. Butov**[3], **Igor I. Baranov**[1], **Dmitry V. Goldshtein**[5], **Galina B. Bolshakova**[4], **Valeria V. Glinkina**[3], **Gennady T. Sukhikh**[1]

**1** Laboratory of Regenerative Medicine, National Medical Research Center for Obstetrics, Gynecology and Perinatology Named After Academician V.I.Kulakov of Ministry of Healthcare of Russian Federation, Moscow, Russian Federation, **2** Histology Department, Peoples Friendship University of Russia (RUDN University), Moscow, Russian Federation, **3** Histology Department, Pirogov Russian National Research Medical University, Ministry of Healthcare of the Russian Federation, Moscow, Russian Federation, **4** Laboratory of Growth and Development, Research Institute of Human Morphology, Moscow, Russian Federation, **5** Laboratory of Stem Cells Genetics, Research Center of Medical Genetics, Moscow, Russian Federation

* elchandrey@yandex.ru

**Data Availability Statement:** All relevant data are within the paper and its Supporting Information files.

## Abstract

Functional and anatomical connection between the liver and the spleen is most clearly manifested in various pathological conditions of the liver (cirrhosis, hepatitis). The mechanisms of the interaction between the two organs are still poorly understood, as there have been practically no studies on the influence exerted by the spleen on the normal liver. Mature male Sprague-Dawley rats of 250–260 g body weight, 3 months old, were splenectomized. The highest numbers of Ki67+ hepatocytes in the liver of splenectomized rats were observed at 24 h after the surgery, simultaneously with the highest index of Ki67-positive hepatocytes. After surgical removal of the spleen, expression of certain genes in the liver tissues increased. A number of genes were upregulated in the liver at a single time point of 24 h, including *Ccne1*, *Egf*, *Tnfa*, *Il6*, *Hgf*, *Met*, *Tgfb1r2* and *Nos2*. The expression of *Ccnd1*, *Tgfb1*, *Tgfb1r1* and *Il10* in the liver was upregulated over the course of 3 days after splenectomy. Monitoring of the liver macrophage populations in splenectomized animals revealed a statistically significant increase in the proportion of CD68-positive cells in the liver (as compared with sham-operated controls) detectable at 24 h and 48 h after the surgery. The difference in the liver content of CD68-positive cells between splenectomized and sham-operated animals evened out by day 3 after the surgery. No alterations in the liver content of CD163-positive cells were observed in the experiments. A decrease in the proportion of CD206-positive liver macrophages was observed at 48 h after splenectomy. The splenectomy-induced hepatocyte proliferation is described by us for the first time. Mechanistically, the effect is apparently induced by the removal of spleen as a major source of Tgfb1 (hepatocyte growth inhibitor) and subsequently supported by activation of proliferation factor-encoding genes in the liver.

**Funding:** The work of AVE, TKF, PAV, MPN, AVL, AVM, IVA, EYK was supported by the Russian Science Foundation (https://rscf.ru/en/, grant number 17-15-01419). The funders had no role in study design, data collection and analysis, decision to publish, or preparation of the manuscript.

**Competing interests:** The authors have declared that no competing interests exist.

## Introduction

The close functional relationship between the liver and the spleen, constituting the so-called hepato-splenic regulatory axis, is emphasized in a number of studies. Anatomical and physiological basis for this axis is provided by portal circulation that connects the two organs, and is cemented by the common barrier and immune functions [1,2]. The mutual influence between the two organs is especially evident in the development of pathological conditions of the liver [1,2].

Splenomegaly and hypersplenism frequently observed in cirrhosis [1] are considered the main causes of cytopenia, notably thrombocytopenia, in cirrhosis patients [3]. Such patients frequently undergo splenectomy with a positive effect [4], which includes alleviation of the portal hypertension, normalization of platelet counts, and notably a reduction in liver fibrosis [4,5]. Positive influence of splenectomy on liver tissue repair has been demonstrated in models of liver transplantation and "small-for-size" liver remnant syndrome [6,7].

Exact mechanisms of the positive effect of splenectomy on liver repair are unknown. It is assumed that under pathological conditions the spleen may act as a source of cytokines and growth factors that affect hepatocyte proliferation, as well as functional activity of the hepatic stellate cells (Ito cells) and liver macrophage populations [8,9].

The interplay between the spleen and the liver was studied in certain pathological processes or corresponding models; under normal conditions, the relationship between the two organs remains poorly understood. This study is focused on the effect of splenectomy on healthy liver.

## Materials and methods

### Animal model

All experimental work involving animals was carried out according to the standards of laboratory practice (National Guidelines No. 267 by the Ministry of Health (Russian Federation), June 1, 2003), and all efforts were made to minimize suffering. The study was approved by the Ethical Review Board at the Scientific Research Institute of Human Morphology (Protocol No. 11, October 4, 2019).

Outbred male Sprague-Dawley rats, body weight 250–260 g, 3 months old, were obtained from the Institute for Bioorganic Chemistry branch animal facilities (Pushchino, Moscow region, Russia). The animals were adapted to the laboratory conditions (23˚C, 12 h/12 h light/dark, 50% humidity, and *ad libitum* access to food and water) for 2 weeks prior to the experiments.

The animals were randomly distributed into two groups (the splenectomy group and the sham-operated control group). The animals (n = 27) were operated at 9:00 to 10:00 a.m. (local time) under general anesthesia with diethyl ether (Medhimprom, Moscow region, Russia; 0.08 ml per liter of chamber volume). Median laparotomy was performed, splenic vein and splenic artery were double-ligated, and the spleen was removed. The anterior abdominal wall was sewn up by layered suturing and the closure was treated with 0.05% chlorhexidine bigluconate. The sham-operated control animals (n = 25) were subjected to median laparotomy subsequently closed by layered suturing with the spleen left intact.

All operated animals (splenectomized and sham-operated) were housed for recovery two per cage in a temperature-regulated room with a 12:12 h light-dark cycle and unlimited access to food and water. Their condition was inspected 4 times a day. As postoperative analgesia, meloxicam (1.0 mg/kg/day) was repeatedly injected into the neck fat pad of all operated animals (splenectomized and sham-operated) for two days after the surgery. Gentamicin (3.0 mg/kg/day) was injected subcutaneously as anti-infective on day 1 after the surgery.

The animals were withdrawn from the experiment in $CO_2$-chamber and weighed at 0 h (before operation), 24 h, 48 h, 3 days or 5 days after the surgery (5–6 animals for each time point). The livers were promptly dissected and weighed (to determine the liver-to-total mass ratio), fixed in 10% buffered formalin for routine H&E staining or frozen in liquid nitrogen for immunochemistry or western blotting (see below).

## Hepatocyte proliferation assessment

Hepatocyte proliferation was assessed by counting mitotic figures or Ki67-positive cells in microscopic images according to a detailed protocol described in our previous work [10]. Hepatocytes were immunostained for Ki67 and CK18 in cryosections. Anti-Ki67 primary antibodies (1:100, Abcam, UK) were topped with fluorescein isothiocyanate (FITC)-conjugated secondary antibodies (1:200, Abcam. Anti-CK18 antibodies (1:100, Abcam) or HNF-4a antibodies (1:100, Santa Cruz) were topped with phycoerythrin (PE)-conjugated secondary antibodies (1:200, Abcam) to produce hepatocyte-specific staining. Cell nuclei were counterstained with 4′,6-diamidino-2-phenylindole (DAPI, Sigma-Aldrich Co LLC). Hepatocyte mitotic index (‰) was calculated for each animal individually by counting mitotic figures per 6,000 hepatocytes. The Ki67 proliferation index (relative counts of Ki67-positive hepatocytes, InKi67) was determined for each animal individually by counting Ki67-positive cells per 3,000 hepatocytes.

## Selective immunostaining of macrophages

Selective immunostaining of liver macrophages in cryosections was implemented with anti-CD68, anti-CD163 or anti-CD206 antibodies (1:100, Abcam) topped with FITC-conjugated secondary antibodies (1:200, Abcam). The nuclei were visualized by DAPI-counterstaining as described previously [10]. The corresponding indexes of macrophage content were determined as relative counts of CD68+, CD163+ and CD206+ cells to the total cell counts in microscopic images.

## Real-time PCR assay

The assay was carried out by a previously described protocol [10]. Total RNA was extracted from the tissues by using RNeasy Plus Mini Kit (QIAGEN) according to the manufacturer's protocol and treated with RNase-free DNase I (Thermo Scientific; 1 U per μg of RNA) to remove traces of genomic DNA. Reverse transcription reactions were set up with the MMLV RT kit (Evrogen, Russia) according to the manual. The reactions (0.5 μg of total RNA each) were carried out at 39˚C for 1 h.

Oligonucleotide PCR primers were custom made by SYNTOL, Moscow, Russia, and used at 0.2–0.4 μmol final concentrations). The reactions were set up in duplicates by using qPCRmix-HS SYBR (Evrogen, Russia) in a reaction volume of 25 μl. Oligonucleotide sequences used in the study are given in Table 1. The amplification was performed in DT-96 real-time thermal cycler (DNA-Technology LLC, Russia) with initial heating at 95˚C for 5 min, followed by 45 cycles of denaturation at 95˚C for 15 s, annealing at 62˚C for 10 s and elongation at 72˚C for 20 s. The relative gene expression values were calculated against *Gapdh* as a reference gene.

## Western-blot analysis of protein representation

Tissue samples (approx. 10 mg) were lyzed with 50 μl of ice-cold RIPA buffer. Subsequent western blot analysis was performed as described previously [11]. After adding 2x loading buffer, the samples were heated at 95˚C for 1 minute, stored at -80˚C until use, and reheated at 95˚C for 1 minute before loading. The proteins were separated by 10% sodium dodecyl sulfate

**Table 1. Oligonucleotide PCR primers used in the study.**

| Target designation | 5′-end primer | 3′-end primer |
|---|---|---|
| Ccnd1 | TGC TTG GGA AGT TGT GTT GG | AATGCCATCACGGTCCCTAC |
| Ccne1 | GAAAATCAGACCGCCCAGAG | CGC TGC AGA AAG TGC TCA TC |
| C-met | AGATTCTGC TGAGCC CATGA | AGA TTC TGC TGA GCC CAT GA |
| Hgf | GGCCATGGTGCTACACTCTT | TTGTGGGGGTACTGCGAATC |
| Il1β | CTGTCTGACCCATGTGAGCT | ACTCCACTTTGGTCTTGACTT |
| Il6 | TACATATGTTCTCAGGGA GAT | GGTAGAAACGGAACTCCAG |
| Il10 | GCCCAGAAATCAAGGAGCAT | TGAGTGTCACGTAGGCTT CTA |
| Tnfa | CCACCACGCTCTTCTGTCTA | GCTACGGGCTTGTCACTCG |
| Hgf | GGCCATGGTGCTACACTCTT | TTGTGGGGGTACTGCGAATC |
| Tgfβ1 | CCGCAACAACGCAATCTATG | AGCCCTGTATTCCGTCTCCTT |
| Tgfβ1R1 | TGCCTGCTTCTCATCGTGTT | TGCTTTTCTGTAGTTGGGAGTTCT |
| Tgfβ1R2 | CCCAAGTCGGTTAACAGCGAT | TGAAGCCGTGGTAGGTGAAC |
| Egf | AGGGTGAACAAGAGGACTGG | TTCACGAATCCTTCCCGACA |
| Egfr | CAGAGCCAACGACTGCCGA | AAATCGCACAGCACCGATCA |
| Arg1 | GGATGAGCATGAGCTCCAAG | GCCAGCTGTTCATTG GCTT |
| iNOs | CGCTGGTTTGAAACTTCTCAG | GGCAAGCCATGTCTGTGAC |
| Gapdh | GCGAGATCCCGCTAACATCA | CCCTTCCACGATGCCAAAGT |

polyacrylamide gel electrophoresis (SDS-PAGE) and transferred from the gel to PVDF membranes by a semi-wet approach using Trans-Blot® Turbo™ RTA Mini LF PVDF Transfer Kit (Bio-Rad Laboratories, Inc., USA). The membranes were blocked with 5% milk in Tris-buffered saline containing 0.1% Tween (TTBS) for 1 h at RT, then stained overnight with primary antibodies to Arg1 (1:200, Santa Cruz), CD206 (1:200, Abcam), HGF (1:200, Abcam), TGFb1 (1:200, Abcam), TNFa (1:200, Abcam), iNOs (1:200, Santa Cruz), Ki67 (1:200, Abcam), cyclin A, cyclin D1, cyclin E1 (1:100, Santa Cruz) or GAPDH (1:200, Santa Cruz), and subsequently with HRP-conjugated secondary antibodies (Bio-Rad). The target proteins were visualized by Novex ECL Kit (Invitrogen™ Thermo Fisher Scientific, USA) in ChemiDoc™ visualization system (Bio-Rad). Optical density of the protein bands was measured in the Image Lab software (Bio-Rad) with GAPDH as a reference protein. The image of full-size membrane after the SDS-PAGE blotting is given in Supplementary Materials.

## Statistical analysis

The data were analyzed by using SigmaStat 3.5 software (Systat Software Inc, USA). Relative gene expression levels, immunophenotype indicators and western blot densitometry measurements were analyzed by Mann-Whitney test; more-than-two-groups comparisons were done using One Way Analysis of Variance with post hoc Holm-Sidak test or ANOVA on ranks with post hoc Tukey test or Dunn's Method. The differences at $p < 0.05$ were considered statistically significant.

## Results

### Liver weight dynamics

None of the animals died of splenectomy or sham intervention. At 24 h after splenectomy, the ratio of liver mass to the total body weight for splenectomized animals was significantly higher than for the control animals (t = 6.609, $p < 0.05$). The increased liver-to-body weight ratio in splenectomized animals was persistent until the end of observation (day 5 after the surgery) (t = 4.112, 4.533, 3.485, respectively, $p < 0.05$) (Fig 1).

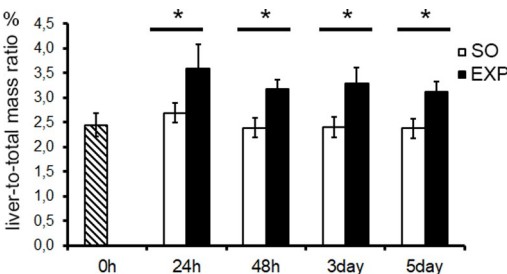

**Fig 1. Dynamics of the liver-to-body weight ratio after splenectomy.** SO, sham operated animals. The data are shown as mean values ± SD with asterisks indicating statistical significance of the differences (as compared with the sham operated control; $p < 0.05$).

## Hepatocyte proliferation

Mitotic figures in hepatocytes of splenectomized animals were observed starting from the 48 h time point, when their frequency was the highest. Dividing hepatocytes were still detectable at 3 and 5 days after splenectomy (Fig 2).

Ki67-positive hepatocytes were detected at 24 h after splenectomy, InKi67 = 2,2 ± 0,4%. Later on, the observable Ki67 expression in the liver ceased (Fig 2). In the anti-Ki67/anti-CK18 or anti-Ki67/anti-HNF4a double-stained cryosections (with cytokeratin CK18 or HNF4a known to be specifically and abundantly expressed in hepatocytes), the vast majority of Ki67-positive cells were identified as hepatocytes. Solitary Ki67$^+$CK18$^-$ cells observed in the vicinity of sinusoid capillaries had the morphology of macrophages or endothelial cells (Fig 3).

Western blot analysis of Ki67 protein content in the liver after splenectomy revealed no difference between splenectomized and sham-operated animals ($p = 0.151$, Fig 2) although an increase in the liver content of this protein in individual splenectomized animals should be noted (Fig 2).

The analysis of gene expression for cyclins D1 and E1 indicated that the expression of *Ccnd1* was upregulated at 24 h, 48 h and day 3 after splenectomy (cyclin D1, q = 4.129, $p = 0.017$, q = 4.229, $p = 0.033$, and q = 3.900, $p = 0.025$ for 24 h, 48 h and day 3, respectively), whereas the expression of *Ccne1* was upregulated at 24 h after the surgery (cyclin E1, q = 2.543, $p$ 0.017).

At the same time, no alterations in the liver content of cyclin A and cyclin D1 proteins were observed ($p = 0.218$  $p = 0.092$), whereas the level of cyclin E1 was increased significantly at 24 h and 48 h after the surgery (t = 4.234, p = 0.003, t = 3.574, p = 0.004, respectively) (Fig 2).

## Immunity- and growth-related gene and protein expression in the normal spleen

The normal splenic tissue showed high expression of *Tgfb1*, low expression of *Hgf*, *Tnfa*, *Il6*, and zero expression levels for *Egf*, *Il10*, *Il12a*, *iNOs* and *Arg1* (Fig 4A).

The data on the protein expression in the spleen was generally consistent with the gene expression profiles. The measurements indicated high levels of TNFa, Tgfb1 and notably CD206, and the lack of iNOS and Arg1 proteins in splenic tissue (Fig 4B).

## Immunity- and growth-related gene expression in the liver after splenectomy

After surgical removal of the spleen, expression of certain genes in the liver tissues increased. A number of genes were upregulated in the liver at a single time point of 24 h, including *Egf* (epidermal growth factor, Q = 4.971, $p < 0.05$), *Tnf* (Tnfa, tumor necrosis factor alpha, Q = 4.069, $p < 0.001$), *Il6* (interleukin 6, Q = 2.818, $p = 0.019$), *Hgf* (hepatocyte growth factor,

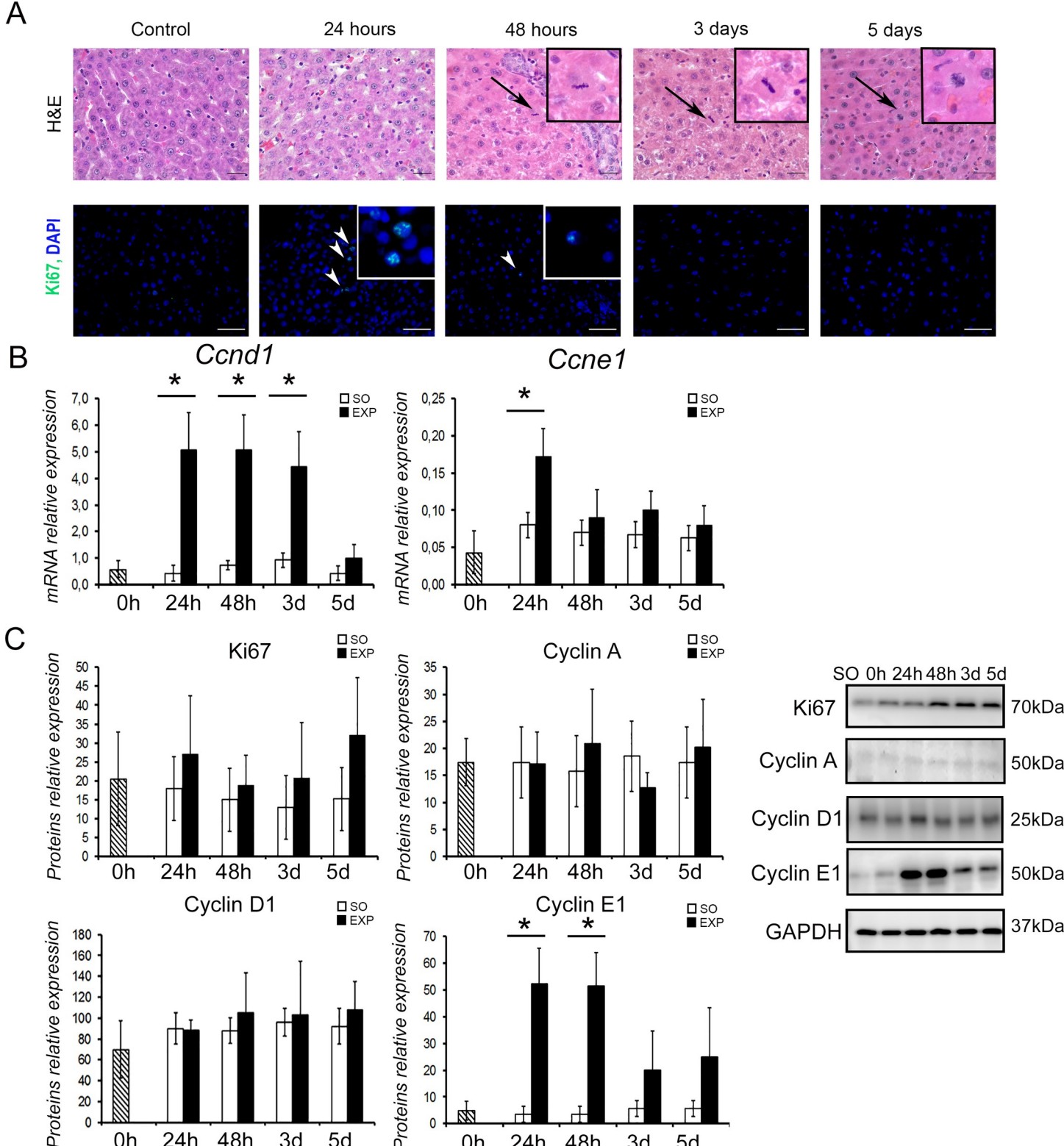

**Fig 2.** Hepatocyte proliferation in the intact liver after splenectomy: (A) Mitotic figures in hepatocytes, hematoxylin-eosin staining (H&E) and Ki67 expression (FITC, nuclei are counterstained DAPI). Control—sham operated animals; Bars, 50 μm;. Arrows indicate mitotic figures in hepatocytes; Arrowheads indicate Ki67+ cells, (B) Expression of *Ccnd1* and *Ccne1* in the liver after splenectomy, (C) Content of Ki67, cyclin A, cyclin D1 and cyclin E1 proteins in the liver after splenectomy.

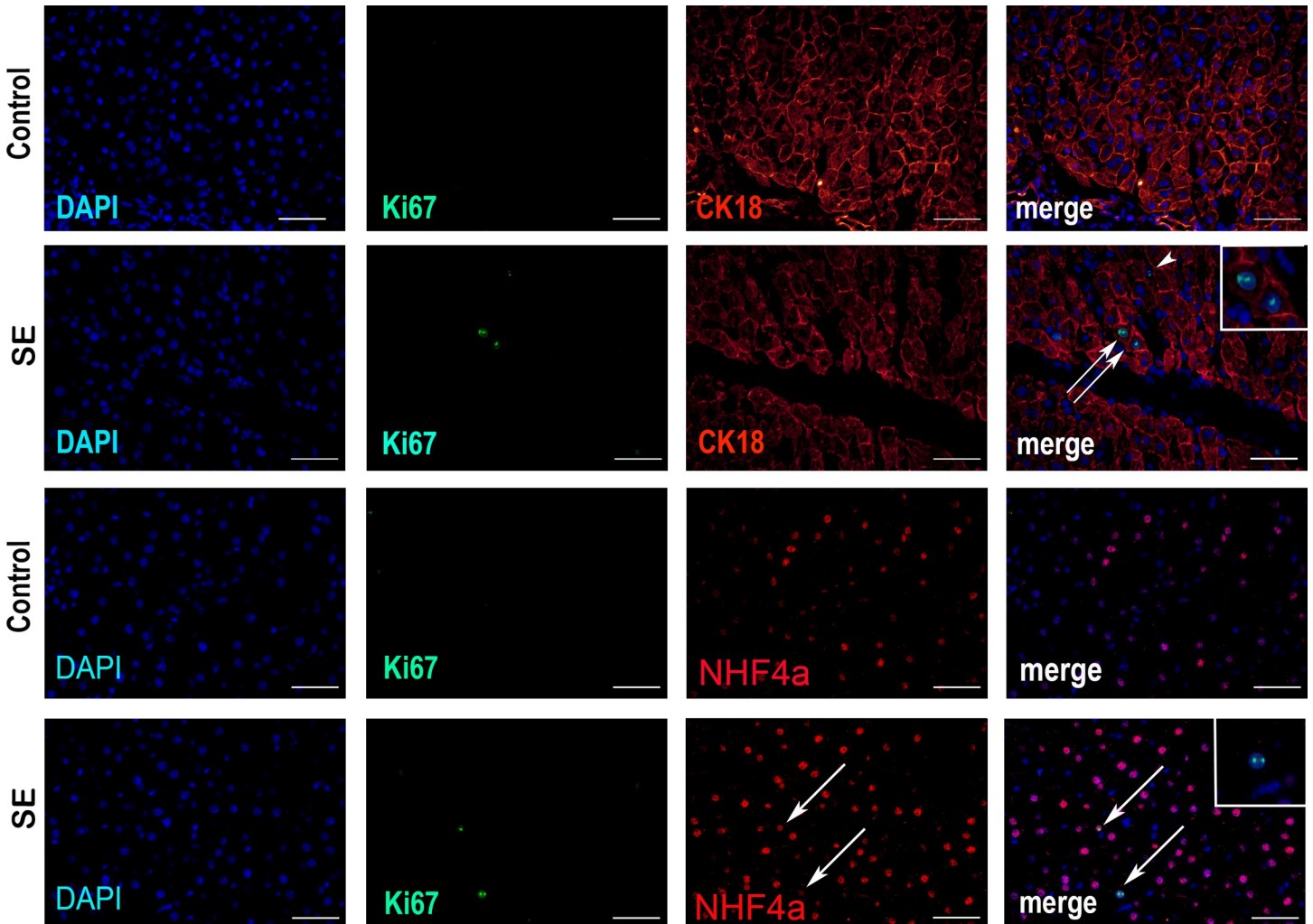

**Fig 3. Splenectomy effects on cell proliferation in the liver.** Cryosections of liver tissue after double anti-Ki67 (FITC) and anti-CK18 (PE) or anti-NHF4a (PE) immunostaining. The nuclei are counterstained with DAPI. Bars, 50 μm. Arrows indicate Ki67+hepatocytes, arrowheads indicate non-parenchymal Ki67+ cells.

Q = 2.53, $p$ = 0.033), *Met* (proto-oncogene c-Met, the hepatocyte growth factor receptor, Q = 3.28, $p$ = 0.004), *Tgfb1r2* (transforming growth factor beta 1 receptor 2, Q = 5.047, $p < 0.05$) and *Nos2* (NO synthase 2, the inducible nitric oxide synthase (iNOs), Q = 3.281, $p$ = 0.004). Expression of *Tgfb1* (transforming growth factor beta 1, Q = 4.435, $p$ = 0.017, Q = 5.129, $p$ = 0.015, and Q = 4.329 $p$ = 0.025), *Tgfb1r1* (transforming growth factor beta 1 receptor 1, Q = 4.665, $p$ = 0.0017, Q = 3.594, $p$ = 0.002, and Q = 3.921, $p$ = 0.025) and *Il10* (interleukin 10, Q = 3.906, $p$ = 0.017, Q = 3.667, $p$ = 0.03, and Q = 3,678, $p$ = 0.025) in the liver was upregulated over the course of 3 days after splenectomy (the $p$-values in parentheses correspond to days 1, 2, and 3, respectively). Expression levels of *Egfr* ($p$ = 0.114) and *Arg1* were constant ($p$ = 0.402) (Fig 5).

### Immunity- and growth-related protein expression in the liver after splenectomy

A statistically significant decrease in the level of Arg1 protein was detected at 24 h and 48 h after splenectomy (Q = 3.570, $p < 0.001$ and Q = 3.323, $p < 0.02$). The level of Tgfb1 was

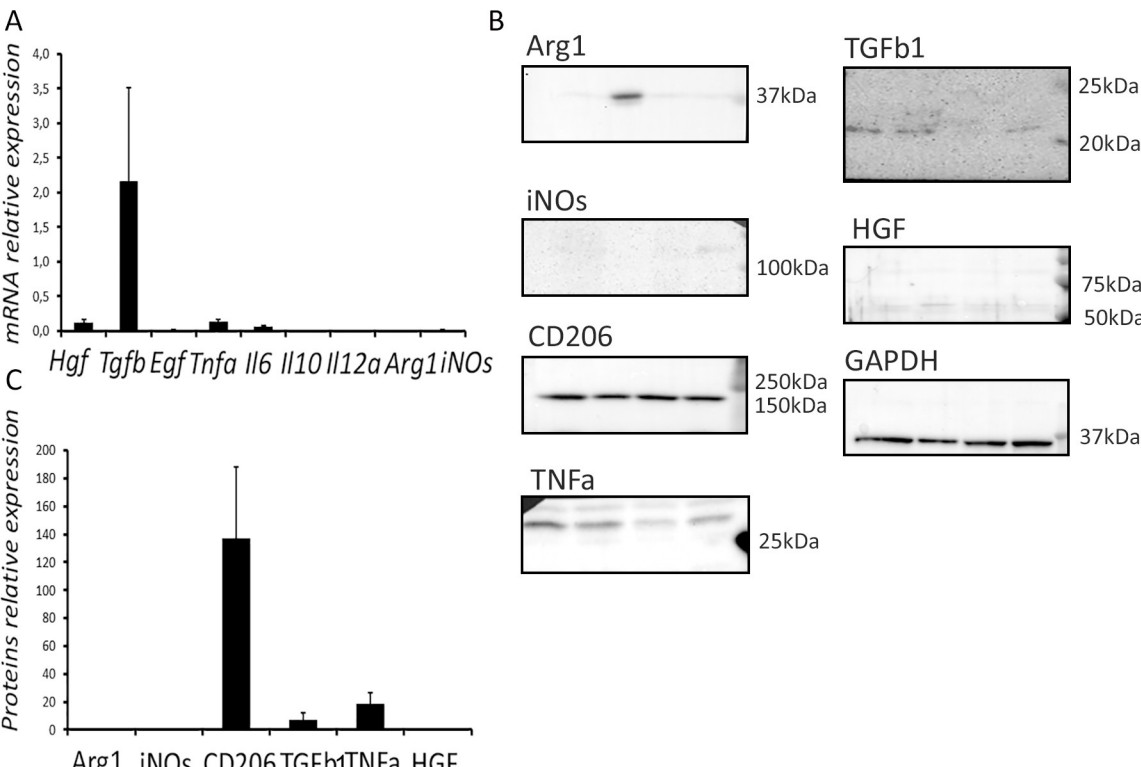

**Fig 4.** Profiles of gene (A) and protein expression (B,C) in the spleen. Visual assessment of the western blot was followed by quantitative densitometry. The data are represented as mean values ± SD.

increased at days 1, 2 and 3 after splenectomy (Q = 3.491, $p$ = 0.017, Q = 3.323 $p$ = 0.033, and Q = 2.706, $p$ = 0.04, respectively), whereas the level of TNFa was decreased significantly at 24 h, 48 h, and 5 days after the surgery (Q = 3.323, $p$ = 0.026, Q = 2.823, $p$ < 0.001, and Q = 3. 760, $p$ = 0.002, respectively). The content of HGF and CD206 proteins in the liver was constant, with a slight tendency towards increase in the CD206 content (Fig 6).

## Population dynamics of liver macrophages after splenectomy

In splenectomized animals, a statistically significant increase in the proportion of CD68-positive cells in the liver (as compared with sham-operated controls) was detectable at 24 h and 48 h after the surgery ($p$ = 0.01 and 0.013, respectively). The difference in the liver content of CD68-positive cells between splenectomized and sham-operated animals evened up by day 3 after the surgery ($p$ > 0.05, Fig 7). No alterations in the liver content of CD163-positive cells were observed in the experiments (Fig 7). A decrease in the proportion of CD206-positive liver macrophages was observed at 48 h after splenectomy as compared with sham-operated controls ($p$ = 0.002, Fig 7).

## Discussion

Development of liver pathologies is often accompanied by 'splenic' symptoms due to the close anatomical and functional connection between the two organs within the so-called hepato-splenic regulatory framework [1,2]. Despite its pronounced character, specific mechanisms of the interaction between the two organs remain understudied.

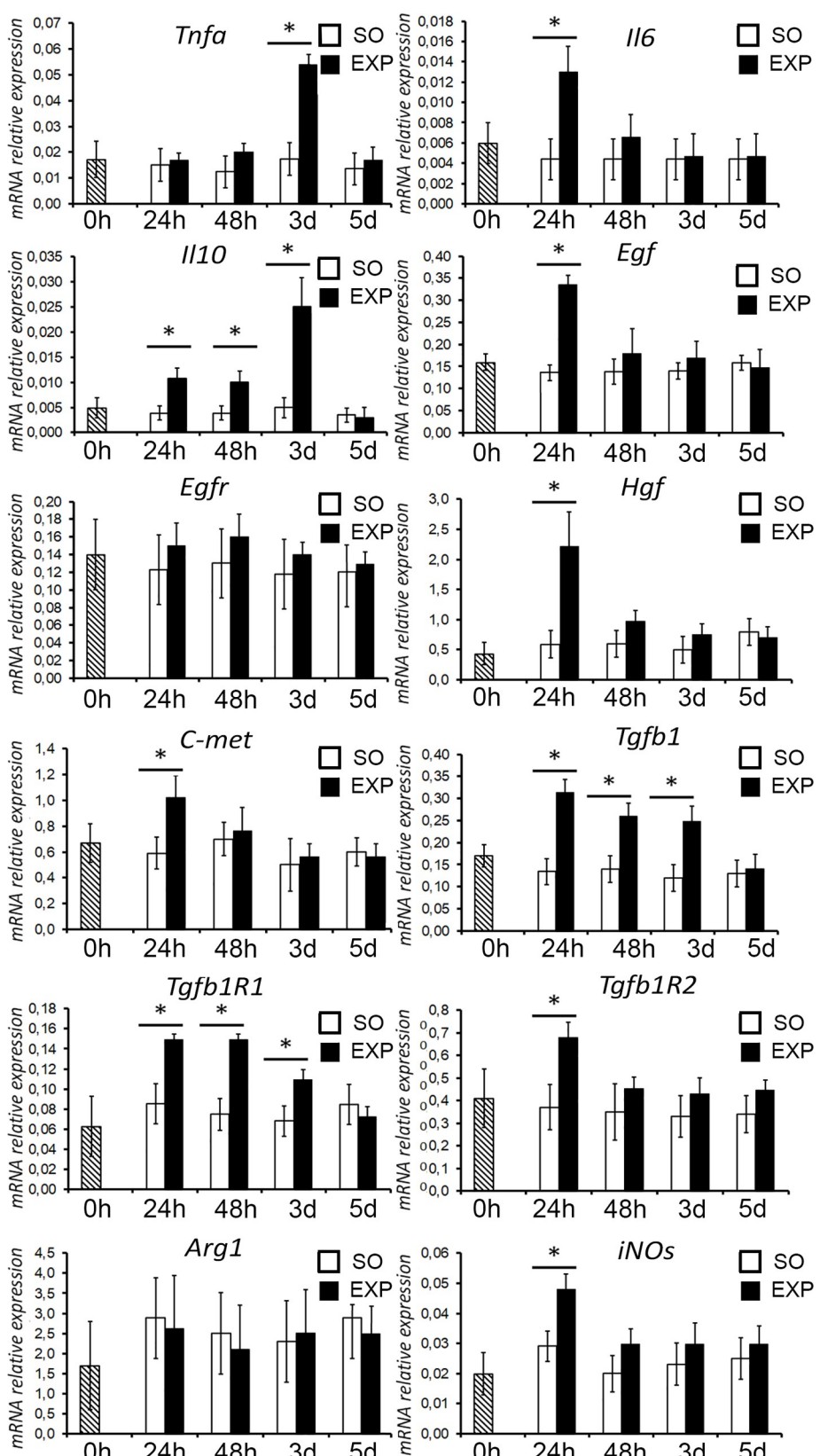

**Fig 5. Splenectomy-associated alterations in gene expression profiles of the liver.** EXP, splenectomized animals; SO, sham-operated animals. The data are shown as mean values ± SD with asterisks indicating statistical significance of the differences (as compared with the sham operated control; $p < 0.05$).

Our research demonstrates that surgical removal of the spleen causes activation of hepatocyte proliferation in normal liver, as evidenced by observations of mitotically dividing hepatocytes, as well as Ki67 expression in liver cells and an increase in the level of the cyclin E1 protein, which stimulates the G1/S-transition [12]. Detectable increase in transcription levels for cyclins D1 was not matched by increased expression of the corresponding protein. The difference in the dynamics of gene expression and the level of synthesis of the corresponding cyclin D1 protein was previously described in a model of liver regeneration after 70% hepatectomy in rats [13]. The authors observed high basic levels of cyclin D1 protein in normal liver. Despite an increase in the level of the corresponding mRNA during regeneration, the protein levels of cyclin D1 remained practically unchanged. This is consistent with the evidence that cyclins D1, as well as cyclin-dependent kinases activated by them, are not ringleaders in the activation and maintenance of hepatocyte proliferation during liver repair [14].

Considering the induction of hepatocyte proliferation by splenectomy, what mechanisms could be responsible for this phenomenon? Two major possible mechanisms should be

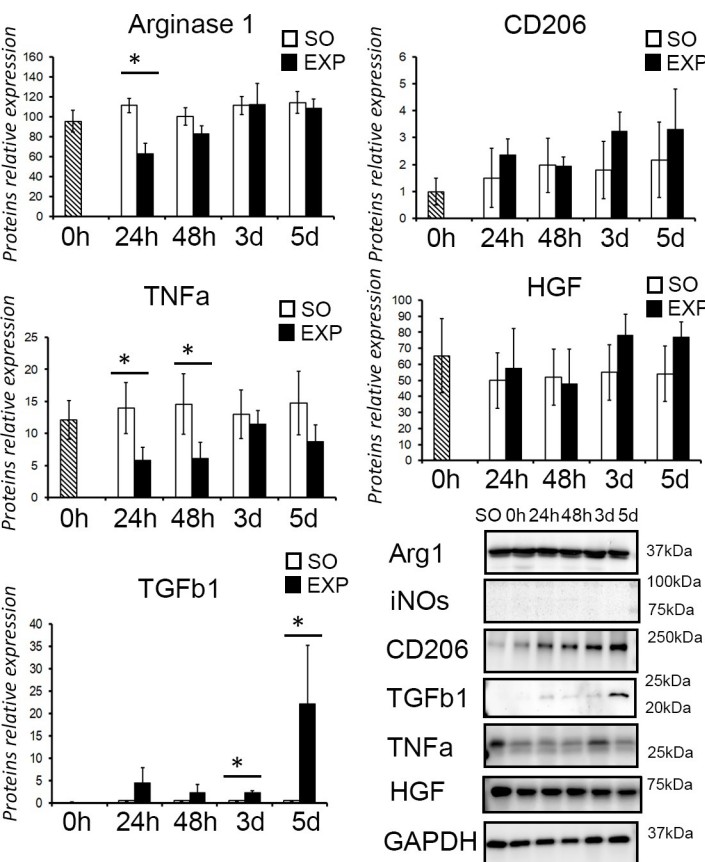

**Fig 6. Splenectomy-associated alterations in protein expression profiles of the liver.** EXP, splenectomized animals; SO, sham-operated animals. Visual assessment of the western blot was followed by quantitative densitometry. The data are shown as mean values ± SD with asterisks indicating statistical significance of the differences (as compared to the sham operated control; $p < 0.05$).

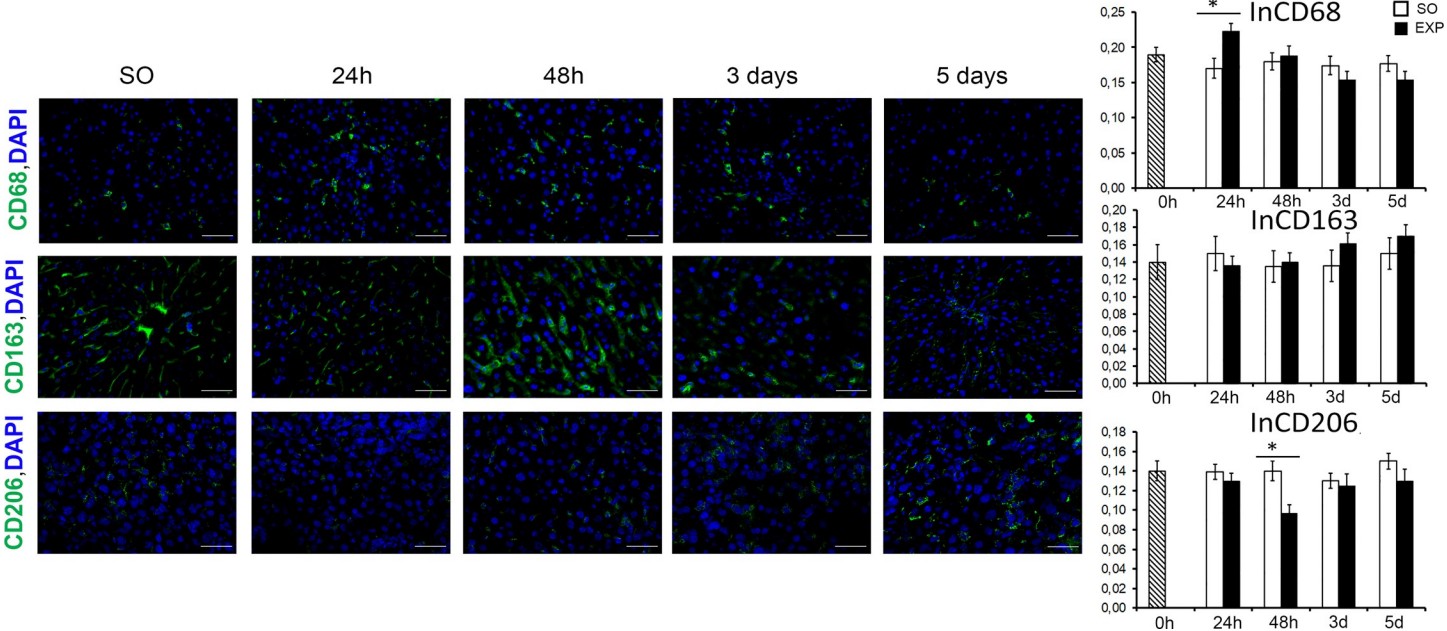

**Fig 7. Splenectomy-associated alterations in the representation of different macrophage populations of the liver, as revealed by immunostaining for the macrophage markers CD68, CD163, and CD206.** The diagrams show dynamic changes in the content of CD68+ (IndexCD68), CD163+ (IndexCD163) and CD206+ (IndexCD206) cells after splenectomy. EXP, splenectomized animals; SO, sham-operated animals. The data are shown as mean values ± SD with asterisks indicating statistical significance of the differences (as compared with the sham operated control; $p < 0.05$).

considered: (i) direct influence of the spleen as a source of stimulatory factors, and (ii) indirect effect of the spleen as a booster of cytokine and growth factor synthesis in the liver.

The first mechanism (direct influence) might be related to the high expression level of *Tgfb1* and the high content of the corresponding protein (transforming growth factor beta 1, Tgfb1) in the spleen. Tgfb1 is known to be a major anti-mitotic factor for hepatocytes [15]. Removal of the spleen apparently cuts its influx to the liver thus loosening the inhibition of hepatocyte proliferation.

In rat model of thioacetamide-induced liver cirrhosis, removal of the spleen promotes a significant decrease in blood levels of Tgfb1, which is beneficial for the liver repair [9]. This result is consistent with the increased content of Tgfb1 in resected splenic tissues of cirrhosis patients (as compared to splenic tissues from patients with normal livers), and colocalization of Tgfb1 with the macrophage marker CD68 revealed by two-color fluorescence immunostaining [8].

The alternative route of hepato-splenic interaction is indirect influence by stimulation of cytokine and growth factor synthesis in the liver itself. The data obtained suggest participation of this route as well, by demonstrating that splenectomy promotes upregulation of a number of cytokine and growth factor genes (*Tnfa*, *Il6*, *Il10*, *Hgf*, *TGfb1*) and their cognate receptors (*C-met*, *TGfb1R1*, *TGfb1R2*) in the normal liver.

The reported findings are partially consistent with results obtained for models of liver pathologies. Several studies have demonstrated that, when the spleen is removed, smaller amounts of pro-inflammatory cytokines (potentially harmful to hepatocytes) enter the regenerating liver; the same is true for Tgfb1, a proliferation blocker [7,16]. The resulting upregulation of Hmox1 protein synthesis in the liver inhibits the activity of TNFa and thus exerts a cytoprotective effect on hepatocytes. In the absence of splenic influence, the synthesis of Tgfb1 and its receptor TGFbRII is reduced, while production of HGF and its cognate receptor c-Met is increased [7,16,17]. In other studies, the stimulating effect of spleen removal on liver

regeneration has been associated with the effect of IL-10, which inhibits hepatocyte proliferation. The *Il10*-deficient mice show higher rates of hepatocyte proliferation following liver resection as compared with normal animals. Due to the fact that, after partial hepatectomy, an increase in IL-10 expression is detected not only in the liver, but also in the spleen, removal of the spleen probably leads to a decrease in the delivery of the hepatocyte proliferation inhibitor IL-10 via the portal vein [18].

Certain discrepancy of the reported data with the previous studies on the role of splenectomy in liver regeneration should be noted. We have observed, for example, that removal of the spleen stimulates expression of certain anti-proliferative genes in liver tissue (*Il10*, *TGfb1*, *Tgfb1R1*, *Tgfb1R2*) along with the genes involved in the activation of hepatocyte proliferation (*Tnfa*, *Il6*, *Hgf*, *C-met*). This relationship between the regulatory factors resembles gene expression profiles in regenerating liver [15,19]. The simultaneous increase in the expression of both mitogens and proliferation blockers is typical for regeneration: it apparently serves as a means for fine control of hepatocyte proliferation [20].

Thus, it can be concluded that the spleen really affects proliferative status of the normal and regenerating liver. The presence of significant indirect component in the structure of this effect raises the question of which cell populations of the liver may mediate it.

A number of studies emphasized the effect of splenectomy on populations of liver macrophages and lymphocytes in cirrhosis. In mice with thioacetamide-induced cirrhosis, splenectomy reduced the degree of fibrotic alterations, stimulated hepatocyte proliferation and increased the numbers of Ly-6C(lo) monocytes and macrophages [21]. At the same time, splenectomy in the rats with induced liver cirrhosis downregulates production of TNFa by liver macrophages (without affecting their numbers), and thereby reduces the degree of liver damage [22].

The data obtained indicate a certain influence on the state of the liver macrophage populations exerted by the spleen. Splenectomy upregulates the expression of macrophage markers (*Tnfa*, *Il6*, *Il10*) and causes a parallel increase in the content of TNFa protein in liver tissues. A simultaneous decrease in the content of Arg1 protein in liver tissues apparently also reflects the influence of splenectomy on the state of liver macrophages.

The increase in the number of CD68-positive cells in the liver after splenectomy, observed by us in this study, may have different possible explanations, e.g. proliferation of Kupffer cells or migration of macrophages to the liver. The analysis of Ki67 and CK18 expression indicates that not only hepatocytes, but also non-parenchymal cells, apparently including macrophages, enter proliferation; this observation is consistent with earlier studies [23]. Increased numbers of liver macrophages after splenectomy may be caused by immigration of macrophages to the liver from other organs. However, such interpretation would require further experimental support.

In addition to the alterations in the abundance of liver macrophages, splenectomy may affect their phenotypes. In mouse model of hepatitis and cirrhosis induced by concanavalin A, splenectomy caused polarization of liver macrophages towards M2 phenotype thus alleviating the condition of the liver [24]. We have also noticed a slight tendency towards an increase in the amount of CD206 protein in the liver after splenectomy. At the same time, we show that splenectomy leads to a decrease in the numbers of CD206-positive cells in the liver, as well as a decrease in concentrations of Arg1 protein (considered as specific marker of the M2 macrophages along with CD206) in liver tissue [25].

Endothelial cells are another cell population of the liver, through which the spleen possibly exerts its influence. Splenectomy accompanied by liver transplantation caused a decrease in the portal pressure and a significant reduction in the severity of endothelial damage in rat

model; it also reduced the levels of apoptosis and inhibited the synthesis of pro-inflammatory cytokines in the liver [6].

Blood flow intensity has been considered as the principal factor in controlling the volume of liver parenchyma; it is thought to influence the condition of the liver through the endothelium of sinusoidal capillaries [26]. Liver resection causes a dramatic increase in the portal blood inflow per functional unit and a concomitant increase in the shear stress per individual cells of the sinusoidal endothelium. In response to the dramatic alteration of hemodynamic conditions, endothelial cells start to produce nitric oxide (NO) which increases the sensitivity of hepatocytes to HGF [27], inhibits Tgfb1 signaling pathway [28], and activates the HGF synthesis [29]. These effects promote massive proliferation of hepatocytes and activate other cell types of the liver. As the liver volume increases, the expansion of sinusoids compensates for the shear stress from portal hypertension and inhibits the liver growth [27,30]. In our model, the increased expression of *iNOs* (*Nos2*) indicates an alteration in the activity of liver endothelium after splenectomy. The increased iNOs activity can lead to overproduction of nitric oxide and concomitant damage to the endothelium [31–33].

It is also worth noting that the stimulating effect of spleen removal on liver repair is not universal and depends on experimental conditions or the nature of pathology. As reported by Babaeva, the lack of spleen has inhibitory effect on regeneration of the liver after massive resections. The inhibitory effect of splenectomy on the compensatory growth of liver tissue does not depend on the time elapsed between the removal of the spleen and subsequent resection of the liver [23]. According to the author, splenectomy causes an increase in the intact liver volume in animal models [23].

To summarize, we report the phenomenon of activated hepatocyte proliferation after splenectomy in rats. Mitotic cycle of hepatocytes is reinforced by splenectomy partially because the spleen normally serves as a source of Tgfb1, which is the main anti-proliferative factor for hepatocytes. The hepatocyte proliferation-stimulating effect of splenectomy can be also due to withdrawal of the splenic influence on the hepatic endothelial cell and macrophage populations, which contributes to the expression of hepatocyte proliferation-activating genes in the liver.

## Supporting information

**S1 Fig. A full-size membrane after SDS-PAGE blotting.** Staining of total proteins with Ponceau S and cutting of the blot for immunostaining are presented. After visualization of the proteins with Ponceau S, the membranes were cut as indicated with a dotted line.
(PDF)

## Acknowledgments

We acknowledge Ekaterina Kirienko for excellent technical assistance. We thank Roman Kholodenko for providing antibodies on revision stage.

## Author Contributions

**Conceptualization:** Andrey V. Elchaninov, Timur Kh. Fatkhudinov, Igor I. Baranov, Dmitry V. Goldshtein, Valeria V. Glinkina.

**Data curation:** Andrey V. Elchaninov, Andrey V. Makarov, Kirill R. Butov, Dmitry V. Goldshtein, Galina B. Bolshakova.

**Formal analysis:** Andrey V. Makarov, Kirill R. Butov.

**Funding acquisition:** Igor I. Baranov, Valeria V. Glinkina, Gennady T. Sukhikh.

**Investigation:** Polina A. Vishnyakova, Maria P. Nikitina, Anastasiya V. Lokhonina, Irina V. Arutyunyan, Evgeniy Y. Kananykhina, Anastasiya S. Poltavets, Kirill R. Butov.

**Methodology:** Polina A. Vishnyakova, Irina V. Arutyunyan, Anastasiya S. Poltavets.

**Project administration:** Timur Kh. Fatkhudinov, Gennady T. Sukhikh.

**Supervision:** Timur Kh. Fatkhudinov, Irina V. Arutyunyan, Galina B. Bolshakova, Valeria V. Glinkina, Gennady T. Sukhikh.

**Validation:** Polina A. Vishnyakova, Galina B. Bolshakova.

**Visualization:** Polina A. Vishnyakova, Anastasiya V. Lokhonina, Evgeniya Y. Kananykhina, Galina B. Bolshakova.

**Writing – Review & Editing:** Andrey V. Elchaninov.

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
