## [Decision Letter · Decision Letter 0]

14 Apr 2020

PONE-D-20-09227

Molecular mechanisms of splenectomy-induced hepatocyte proliferation

PLOS ONE

Dear Dr. Elchaninov,

Thank you for submitting your manuscript to PLOS ONE. After careful consideration, we feel that it has merit but does not fully meet PLOS ONE’s publication criteria as it currently stands. Therefore, we invite you to submit a revised version of the manuscript that addresses all issues raised by the reviewer.

We would appreciate receiving your revised manuscript by May 29 2020 11:59PM. To enhance the reproducibility of your results, we recommend that if applicable you deposit your laboratory protocols in protocols.io, where a protocol can be assigned its own identifier (DOI) such that it can be cited independently in the future. For instructions see: http://journals.plos.org/plosone/s/submission-guidelines#loc-laboratory-protocols

We look forward to receiving your revised manuscript.

Kind regards,

Matias A Avila, Ph.D.

Academic Editor

PLOS ONE

Journal Requirements:

Reviewers' comments:

Reviewer's Responses to Questions

**Comments to the Author**

1. Is the manuscript technically sound, and do the data support the conclusions?

Reviewer #1: Partly

2. Has the statistical analysis been performed appropriately and rigorously? 

Reviewer #1: No

3. Have the authors made all data underlying the findings in their manuscript fully available?

Reviewer #1: Yes

4. Is the manuscript presented in an intelligible fashion and written in standard English?

Reviewer #1: No

5. Review Comments to the Author

Reviewer #1: In this study Andrey Elchaninov and colleagues investigated the outcomes and molecular mechanisms of splenectomy-induced hepatocyte proliferation. The findings are of interest, however lacking sufficient novelty and the general organisation of the paper is not satisfactory to my point of view.

Several points should be addressed:

1. All operated animals received Meloxicam (1.0 mg/kg/day) and additionally gentamicin (3.0 mg/kg/day). Did animals from the sham group received the same treatment?

2. Information about the age of rats should be included.

3. Figure 1

- Liver/body weight ratio has to be converted into the form of a percentage (%)

- The liver/body weight ratio before operation (0h time point) has to be presented for both groups

- Did authors check the hepatosomatic ratio at later time points? 10-14 days after splenectomy?

4. Figure 2

- The quality of H&E and especially Ki67 figures is very low.

Moreover, the chosen areas in the liver sections are not representative. Such as, only the control and 24 h H&E images include the hepatic area with central vein and therefore can’t be compared to the rest of images.

-The ki67positive cells are not visible.

-The quantification of the images should be presented as diagram.

5. The general structure of the paper has to be changed. Figure 2 has to be reorganised. Cell cycle related qPCR has to be included into figure 2. Moreover, authors should perform PCNA and cyclin D Western blots.

6. Figure 3. I don’t recommend to use CK18 as general hepatocytes marker. Instead, the authors can use HNF4α staining for hepatocytes. Another option: authors can perform IHC Ki67 staining. With detail analysis of IHC Ki67 images the proliferating hepatocytes (relatively big, brown round nuclei) can be easily distinguished from nonparenchymal cells.

7. The quality of CD68 IF staining is very low. May be include F4/80 staining can be another option.

8. Please include 0h (before operation) controls in all qPCR

9. The English in the presented manuscript is not of publication quality and require major improvement

6. PLOS authors have the option to publish the peer review history of their article (what does this mean?). If published, this will include your full peer review and any attached files.

Reviewer #1: No

---

## [Author Response · Author response to Decision Letter 0]

4 May 2020

Dear reviewers, 

Thank you for the detailed and in-depth analysis of the submitted manuscript. We have tried to answer all your questions in detail and to the point. 

1. All operated animals received Meloxicam (1.0 mg/kg/day) and additionally gentamicin (3.0 mg/kg/day). Did animals from the sham group received the same treatment?

Yes; we specify this in the revision.

2. Information about the age of rats should be included.

The corresponding section has been corrected. 

3. Figure 1 - Liver/body weight ratio has to be converted into the form of a percentage (%)

- The liver/body weight ratio before operation (0h time point) has to be presented for both groups

- Did authors check the hepatosomatic ratio at later time points? 10-14 days after splenectomy?

The corresponding figure has been corrected. 

The ratio of liver mass and body weight of experimental rats at later time points (10-14 days after the surgery) was not examined. The observed hepatocyte proliferation was transient, as indicated by the lack of Ki67-positive hepatocytes after day 1, and the ratio of liver mass to body weight for the operated rats remained constant, similarly with the control animals. 

4. Figure 2

- The quality of H&E and especially Ki67 figures is very low.

Moreover, the chosen areas in the liver sections are not representative. Such as, only the control and 24 h H&E images include the hepatic area with central vein and therefore can’t be compared to the rest of images.

-The ki67positive cells are not visible.

-The quantification of the images should be presented as diagram.

All original illustrations (charts and microphotographs) were uploaded to the Editorial Manager in high quality with a resolution of 300 dpi. The original microphotographs clearly showed the specific staining, as well as characteristic features of liver microanatomy. Apparently, a decrease in the image quality occurred upon subsequent conversion of the image for the Reviewer’s copy of the manuscript. In the revised manuscript, the photographs of H&E stained sections of the liver are replaced with more representative images. 

5. The general structure of the paper has to be changed. Figure 2 has to be reorganised. Cell cycle related qPCR has to be included into figure 2. Moreover, authors should perform PCNA and cyclin D Western blots.

Figure 2 has been corrected. The western blot analysis was performed for cyclins D1 and E. At the same time, we prefer Ki67 as a proliferation marker for its specificity, since it has been reported to produce fewer false positive signals compared to PCNA [https://www.ncbi.nlm.nih.gov/pubmed/14623936]. 

6. Figure 3. I don’t recommend to use CK18 as general hepatocytes marker. Instead, the authors can use HNF4α staining for hepatocytes. Another option: authors can perform IHC Ki67 staining. With detail analysis of IHC Ki67 images the proliferating hepatocytes (relatively big, brown round nuclei) can be easily distinguished from nonparenchymal cells.

CK18 is a well-established hepatocyte marker [https://www.ncbi.nlm.nih.gov/pubmed/21420381]; for this reason, we preserved the original illustrations supplementing them with microphotographs showing the combined expression of HNF4α and Ki67 in hepatocytes. 

7. The quality of CD68 IF staining is very low. May be include F4/80 staining can be another option.

The corresponding figure has been corrected. We selected more representative microphotographs that appropriately illustrate CD68 expression in the liver. F4/80 is a well-established macrophage marker indeed, although it has been predominantly characterized for mice and partially for humans, and not for rats [https://www.ncbi.nlm.nih.gov/pubmed/28455709]. 

8. Please include 0h (before operation) controls in all qPCR

The corresponding section and figures have been corrected.

9. The English in the presented manuscript is not of publication quality and require major improvement

The proofreading and text editing have been performed.

Yours faithfully,

Dr. Andrey Elchaninov 

Corresponding Author

Dr Andrey Elchaninov, Laboratory of Regenerative Medicine, National Medical Research Center for Obstetrics, Gynecology and Perinatology Named after Academician V.I. Kulakov of Ministry of Healthcare of Russian Federation, Moscow, Russia, 4 Oparina Street, Moscow 117997, Russia E-mail: elchandrey@yandex.ru; phone: +7 (916) 8885292

---

## [Editor Report · Decision Letter 1]

13 May 2020

Molecular mechanisms of splenectomy-induced hepatocyte proliferation

PONE-D-20-09227R1

Dear Dr. Elchaninov,

We are pleased to inform you that your manuscript has been judged scientifically suitable for publication and will be formally accepted for publication once it complies with all outstanding technical requirements.

With kind regards,

Matias A Avila, Ph.D.

Academic Editor

PLOS ONE
---

## [Editor Report · Acceptance letter]

15 May 2020

PONE-D-20-09227R1 

Molecular mechanisms of splenectomy-induced hepatocyte proliferation 

Dear Dr. Elchaninov:

I am pleased to inform you that your manuscript has been deemed suitable for publication in PLOS ONE. Congratulations! Your manuscript is now with our production department. 

With kind regards,

on behalf of

Dr Matias A Avila 

Academic Editor

PLOS ONE